# Identification and Molecular Characterization of a Divergent Asian-like Canine Parvovirus Type 2b (CPV-2b) Strain in Southern Italy

**DOI:** 10.3390/ijms231911240

**Published:** 2022-09-24

**Authors:** Giorgia Schirò, Francesco Mira, Marta Canuti, Stefano Vullo, Giuseppa Purpari, Gabriele Chiaramonte, Santina Di Bella, Vincenza Cannella, Vincenzo Randazzo, Calogero Castronovo, Domenico Vicari, Annalisa Guercio

**Affiliations:** 1Istituto Zooprofilattico Sperimentale della Sicilia “A. Mirri”, Via Gino Marinuzzi 3, 90129 Palermo, Italy; 2Department of Pathophysiology and Transplantation, Università degli Studi di Milano, Via Francesco Sforza 35, 20122 Milano, Italy

**Keywords:** CPV-2, *Carnivore protoparvovirus 1*, dog, parvovirus, Asia, Europe, phylogenetic analysis, molecular typing

## Abstract

Canine parvovirus type 2 (CPV-2) is an infectious agent relevant to domestic and wild carnivorans. Recent studies documented the introduction and spread of CPV-2c strains of Asian origin in the Italian canine population. We investigated tissue samples from a puppy collected during necropsy for the presence of viral enteropathogens and all samples tested positive only for CPV-2. The full coding sequence of a CPV-2b (VP2 426Asp) strain was obtained. This virus was related to CPV-2c strains of Asian origin and unrelated to European CPV-2b strains. The sequence had genetic signatures typical of Asian strains (NS1: 60Val, 545Val, 630Pro; VP2: 5Gly, 267Tyr, 324Ile) and mutations rarely reported in Asian CPV-2b strains (NS1: 588N; VP2: 370Arg). Phylogenetic analyses placed this strain in well-supported clades, including Asian CPV-2c-like strains, but always as a basal, single-sequence long branch. No recombination was observed for this strain, and we speculate that it could have originated from an Asian CPV-2c-like strain that acquired the 426Asp mutation. This study reports the first evidence of an Asian-like CPV-2b strain in Italy, confirming the occurrence of continuous changes in the global CPV-2 spread. Since positive convergent mutations complicate data interpretation, a combination of phylogenetic and mutation pattern analyses is crucial in studying the origin and evolution of CPV-2 strains.

## 1. Introduction

Canine parvovirus type 2 (CPV-2) is taxonomically included in the species *Carnivore protoparvovirus 1*, within the genus *Protoparvovirus* (family *Parvoviridae*, subfamily *Parvovirinae*) [1]. CPV-2 is a small, non-enveloped, single-stranded DNA virus whose genome consists of an approximately 5200 nucleotide (nt) long DNA molecule, containing two gene cassettes encoding, through alternative splicing of the same mRNAs, for non-structural (NS1–NS2) and structural (VP1–VP2) proteins [2,3].

The original CPV-2 type emerged in the late 1970s as a host variant of a feline panleukopenia virus (FPLV), or FPLV-like parvovirus, and rapidly spread worldwide, causing an acute enteric disease and leucopenia in canine species [4,5]. This original viral strain was replaced by the genetic variant known as CPV-2a (87Leu, 101Thr, 305Tyr), from which a further two antigenic variants (namely CPV-2b, and CPV-2c) originated. The three antigenic variants, which are all included in the same phylogenetic clade (CPV-2a clade), have been reported to circulate at different rates, depending on the time or the country of sample collection. They also regained the ability to infect feline species [5,6]. These three CPV-2 variants differ from each other by the amino acids at residue 426 (CPV-2 and CPV-2a: Asn; CPV-2b: Asp; CPV-2c: Glu) in the major capsid protein VP2 [2]. A further substitution at residue 297 of the VP2 protein (Ser to Ala) was detected for the first time in 1990, and this mutation is now used as a genomic marker of the so-called “new CPV-2a/2b” variants, currently predominant among strains circulating in dogs [7]. However, although possibly antigenically relevant [2], virus typing based on the analysis of these sole residues does not reflect the viral evolution or the phylogenetic relationships between strains as viruses of the same antigenic type often do not cluster together [8,9]. This is likely due to selection pressure forces inducing the emergence of the same mutations multiple times during the evolutionary history of this virus [8].

Currently, despite the widespread use of effective vaccines, CPV-2 is still a worldwide concern as it is responsible for a severe and often fatal disease in domestic and wild carnivorans [10]. In the last decade, several studies have analyzed the spread of CPV-2 in domestic and wild carnivorans in Italy [11,12] and documented the introduction and rapid diffusion of CPV-2c strains of Asian origin (Asian CPV-2c-like strains) in the Italian canine population [13,14]. These strains, which have now been detected worldwide [15,16,17], are characterized by a specific set of mutations in both structural and non-structural proteins and are phylogenetically related to each other. 

The occurrence of introductions and subsequent spread of divergent variants in new geographical areas supports the need for systemic epidemiological surveys to evaluate the circulation and evolution of CPV-2. This study reports the identification, genomic characterization, and sequence analysis of a divergent CPV-2b strain related to Asian CPV-2c-like strains detected in a stray puppy in Italy.

## 2. Results

### 2.1. Clinical Case

In January 2022, a litter of fifty-day-old mixed breed puppies displayed clinical signs of gastroenteritis. The five stray puppies were recovered from Sciacca (Sicily region) in southern Italy for necessary therapies. After a few days of gastroenteric signs (anorexia, vomit, bloody diarrhea and fever), and despite hospitalization and intensive care, all puppies died, and one was subjected to necropsy for diagnostic purposes. 

At necropsy, hyperemia of the gastric mucosa, catarrhal fluids in the stomach, hemorrhage of the serous membrane of the small intestine, congestion and enlargement of mesenteric lymph nodes, pulmonary oedema with marginal petechiae and necrosis were observed. Tissue samples (heart, lungs, spleen, liver, intestine, kidney) were collected for virological investigation and screened for various enteropathogens. All tissue samples tested positive for CPV-2 by conventional PCR and negative for canine distemper virus, canine adenovirus, canine coronavirus, norovirus and rotavirus by real-time (RT)-PCR.

### 2.2. Identification, Genetic Characterization, and Phylogenetic Analysis of a Divergent Strain

From one of the samples, a 4269 nt-long parvoviral sequence was obtained, and the strain (CPV-2b_IZSSI_2022PA2773) was characterized as CPV-2b on the basis of the Asp at residue 426 of the VP2 protein. The strain also belonged to the group currently referred to in the literature as “new CPV-2b”, based on the presence of Ala at residue 297 of VP2. The nearly complete genome sequence of CPV-2b_IZSSI_2022PA2773, which included the full coding region and only lacked the terminal untranslated ends, showed the highest nucleotide identities with CPV-2c strains found in samples collected in 2018 from Nigerian dogs (99.60–99.58%), in 2017 from dogs from Shanghai (99.58%), in 2017 from a Chinese dog and from a dog imported to Italy from Bangkok, and in 2016 from a cat from Bangkok (99.55%) (Appendix A). All these strains belonged to the Asian CPV-2c-like clade (see below).

The analysis of the CPV-2b_IZSSI_2022PA2773 NS1 gene sequence showed the highest nucleotide identity (99.75%), with the NS1 of CPV-2 strains collected from dogs in China in 2014, 2017 and 2019, in Vietnam in 2017, in Italy in 2017 and 2018, and in Nigeria in 2018 (Appendix A). Sequence analysis revealed the presence of NS1 in substitutions observed in Asian CPV-2c-like strains (60Val, 544Phe, 545Val, 630Pro) (Table 1). NS1 residues 60Val e 544Phe were also recorded in the FPV strain TZ-FPV-143 (acc.nr. MZ836440) collected in China in 2020. Additionally, the amino acid (aa) substitution S588N was observed in CPV-2b_IZSSI_2022PA2773 and this was also reported at a low rate (1.4%) in less-related CPV-2 sequences (99.40–99.10% identity) with a heterogeneous origin (Mexico in 2015, acc.nr. MT448706; China in 2019, MZ836307 and MZ836309; USA in 2019, MN451684; Iran in 2020, MW653252; vaccinal strains from Japan, LC270891 and LC270892) and in four FPLV sequences collected from two cats, a giant panda and a red panda from China in 2007, 2016 and 2020, respectively (MG924893, EF988660, MZ357119, and MW331496).

The analysis of the VP2 gene of the CPV-2b_IZSSI_2022PA2773 sequence showed the highest nucleotide identity (99.50%), with the VP2 gene sequences of CPV-2c strains collected in Asia (China in 2017, 2019 and 2020; Thailand in 2016 and 2020; Indonesia in 2013; South Korea and Vietnam in 2017), Europe (Italy in 2017 and Romania in 2019) and Africa (Nigeria in 2018), and of CPV-2b strains collected in Australia (2015–2016) and Thailand (2010) (Appendix A). The presence of the aa residue characteristics of various antigenic types of CPV-2 strains of Asian origin (5Gly, 267Tyr, 324Ile, 370Arg) were observed (Table 1). However, Gly at residue 5 was shared mainly with CPV-2c strains of Asian origin, with the exception of three CPV-2b strains from China (collected in 2018, acc.nr. MK268683, nt identity 99.48%; in 2017, MK518011, 99.37%; in 2018, MN119560, 99.14%), among others, and twenty-six CPV-2b strains collected in Australia. Furthermore, residue 370Arg was shared mainly with CPV-2c strains and a few CPV-2a strains, as well as only one CPV-2b strain of Asian origin. These characteristics make this strain closer to Asian-like CPV-2c strains, rather than CPV-2b strains. Interestingly, three of these CPV-2b strains retaining the 5Gly and/or 370Arg residues (MK518011, MN119560, MK518007) originated from the Anhui province in China.

Comparing the obtained sequences to CPV-2b/2c strains [9,18] and to Asian CPV-2c-like strains [13,14,18] recently collected in the same Italian region, the VP2 gene of CPV-2b_IZSSI_2022PA2773 showed a higher nucleotide identity to Asian CPV-2c-like strains (99.60–99.37%) than to other locally circulating CPV-2b (98.80–98.63%) or CPV-2c (98.97–98.51%) strains.

Finally, to investigate the phylogenetic relationships between the strain identified in this study and other globally circulating viruses, three maximum likelihood trees were built, which included all complete genomes, full NS1, or full VP2 currently available in GenBank. In each phylogenetic tree, CPV-2b_IZSSI_2022PA2773, which was not detected as recombinant in any of the used alignments, was included in large and highly supported clusters that included CPV-2a (N = 4), CPV-2b (N = 3) and CPV-2c (N = 491) strains identified in various countries, including Italy (Figure 1 and Appendix A). 

These clusters, highlighted in red in Figure 1, corresponded to the Asian CPV-2c-like clade as shown in Appendix A. Specifically, the amino acid sequence analysis performed with all available sequences showed that the set of mutations that characterizes Asian CPV-2c-like strains (Table 1) is unique to this clade, although a few sequences with mutations at key sites in this clade existed (Appendix A). More specifically, the NS1 mutation combination 60Val/544Phe/545Val, as well as a Pro at residue 630 of NS1, existed only in this clade. Additionally, the amino acid combination in VP2 267Tyr/297Ala/324Ile/370Arg/440Thr, observed in 98.6% of the strains within this clade, was only detected in this clade and in another nine CPV-2c sequences phylogenetically far from Asian CPV-2c-like strains. 

Nonetheless, the study strain was never included in any supported sub-clusters together with other strains and it always formed long single-sequence branches, which were somewhat close to strains identified in Australian dogs. These results indicate that the geographical origin of this CPV-2b strain, which is the most similar to but still divergent from Asian CPV-2c-like strains, remains uncertain.

## 3. Discussion

Despite new CPV-2a/2b variants being the prevalent strains in Asia for many years [19,20], a CPV-2c variant with specific genetic markers has been sporadically detected in the continent since 2009, and its detection rate has increased through the years, becoming the predominant CPV-2 variant in some Asian countries since 2018 [21,22,23]. The introduction of an Asian CPV-2c strain in Europe through dog importation in 2017, and its rapid spread in the following years, both in domestic dogs and wild carnivorans, has been documented in Italy [12,13,14]. In recent years, the Asian CPV-2c variant was reported in another European country [15], as well as in Africa [16] and North America [17], supporting its progressive worldwide spread in the canine population. Nonetheless, the CPV-2a/2b strains that have been prevalent for many years in Asia have not been reported in Europe as of yet. 

This study reports the detection of a CPV-2b strain displaying genetic signatures typical of Asian CPV-2c-like viruses in an Italian stray puppy. Compared to European CPV-2 strains, this CPV-2b strain exhibits specific amino changes in NS1 (Ile60Val, Tyr544Phe, Glu545Val, Leu630Pro) and VP2 (Ala5Gly, Phe267Tyr, Tyr324Ile, Gln370Arg) proteins, some of which have been observed in CPV-2 strains identified in Asia since 2013 [24]. 

Interestingly, this CPV-2b strain showed two VP2 amino acid residues (5Gly and 370Arg) observed mainly in Asian CPV-2c strains, and only less frequently described in CPV-2b variants. The recombination analysis, based on the largest database available at the moment, excluded that this strain originated after an event of recombination between currently known CPV-2 variants; therefore, this amino acid signature could be the result of convergent evolution. Specifically, it is possible that the strain detected in this study is a CPV-2b strain that acquired mutations typical of CPV-2c viruses or, given its phylogenetic relatedness and high similarity to Asian CPV-2c-like strains, it more likely originated from a CPV-2c strain that acquired the 426Asp mutation. This could also be the case for the other CPV-2b and CPV-2a strains detected in the Asian CPV-2c-like clade. Indeed, mutations at this site occur frequently and it has been shown that they have emerged multiple times throughout the evolutionary history of CPV-2 [8]. Although this strain is phylogenetically close to viruses identified in Australia, phylogenetic analyses did not precisely place it in supported clades including any other known strains; therefore, we unfortunately cannot make a hypothesis about the geographical origin of this strain. Due to the wide distribution of Asian CPV-2c-like strains in Europe [14,15,25], the possibility that this Asian-like CPV-2b strain originated from another European or a non-Asian country and was then introduced in Italy cannot be excluded. Nonetheless, the identified mutations could represent a specific molecular marker for perspective future molecular analyses to assess the spread of this virus. Additionally, VP2 is a major capsid protein and is the target of the host’s antibody response [26]. Further studies are necessary to elucidate any potential advantage of this CPV-2b strain. Due to the coexistence of genetically divergent strains in the same geographical environment, it would also be interesting to monitor over time the distribution, change in relative frequency and evolution of CPV-2 variants in Europe [14]. 

Another amino acid substitution of interest was observed in the NS1 protein: the amino acid Asn at position 588 has been observed in a minority of analyzed sequences with a heterogenous origin. Interestingly, this residue has also been observed in two FPLV strains from domestic cats (acc.nr. MG924893; EF988660, [27]), and from giant and red pandas (MZ357119 and MW331496, [28]). These changes could also be the result of convergent evolution. 

Both CPV-2a/2b and CPV-2c variants of Asian origin have been infrequently observed in cats in Asia [29,30] and in wolves in Italy [12]. As these reports were based on partial or complete VP2 gene sequences and, considering the key NS1 genetic features of the investigated CPV-2b strains, further molecular surveys coupled with complete genome sequencing could also be suggested for different domestic and wild animals from Italy to elucidate the spread of this CPV-2 variant also in domestic cats or feral carnivorans.

The limited availability of updated NS genes or complete CPV-2 genome sequences, particularly for the CPV-2b variants detected in Asia, limited the power of our analyses and prevented definitive conclusions about the origin and evolution of this CPV-2b strain. Indeed, most studies on *Carnivore protoparvovirus 1* are still focused on the VP2 gene sequence and contributions that also include the genetic analysis of the nonstructural genes are in the minority, limiting the availability of sequence data for comparison [9,31]. Additionally, the presence of sites under positive-selection pressure and of convergent mutations makes data interpretation more complicated, and this highlights how a combination of phylogenetic and mutation pattern analyses is crucial to study the origin and evolution of CPV-2 strains. In fact, the strain identified in this study could be antigenically classified as CPV-2b, but it was phylogenetically part of the Asian CPV-2c-like clade with which it shared several mutations in both structural and non-structural proteins. This led us to speculate that this strain originated from a CPV-2c strain that acquired the antigenic characteristic of a CPV-2b strain, and it is possible that similar conclusions could be made about other CPV-2b and CPV-2a strains with a mutation pattern similar to Asian CPV-2c-like strains. This further highlights, as we previously observed [9], how a classification based on mutations at one single site is unreliable to study the spread of CPV-2, but it indicates that this could be possible by using a set of mutations throughout the whole genome.

## 4. Materials and Methods

### 4.1. Nucleic Acid Extraction and Viral Screening

Organ homogenates were obtained as previously described [32]. Total DNA and RNA were extracted from homogenates using the DNeasy Blood & Tissue Kit (Qiagen S.p.A, Milan, Italy) and the QIAamp Viral RNA Mini Kit (Qiagen S.p.A.), respectively, according to the manufacturer’s instructions. 

Extracted DNA/RNA was subjected to a set of PCRs and RT-PCRs useful to evaluate the presence of specific viruses: CPV-2 (as described in [7]), canine distemper virus (as described in [33]), canine coronavirus [34], canine adenovirus type 1 and type 2 [35] and norovirus [36]. Each amplicon was analyzed by electrophoresis on a 3% agarose gel, supplemented with ethidium bromide. A RT-PCR assay was performed to evaluate the presence of rotavirus RNA [37].

### 4.2. Sequence, Phylogenetic, and Recombination Analysis

Sequencing of the near complete genome, encompassing both ORFs encoding for NS and VP genes, was carried out using primer pairs developed by Pérez et al. [38], as previously described [9]. Amplicons were purified with Illustra™ GFX™ PCR DNA and the Gel Band Purification Kit (GE Healthcare Life Sciences, Amersham, Buckinghamshire, UK), and were submitted to BMR Genomics srl (Padova, Italy) for direct Sanger sequencing with additional internal primers, as previously described [9]. Sequences were assembled according to an overlapping strategy and analyzed using Geneious Prime 2022.0.2 (Biomatters, San Diego, CA, USA). Assembled nucleotide sequences were submitted to BLASTn [39] to search for related sequences in public databases (https://blast.ncbi.nlm.nih.gov/Blast.cgi, accessed on 10 March 2022). Key amino acid variations were deduced by comparing the analyzed sequence with the whole dataset, and they were synthetically depicted in Table 1 by including available nearly complete CPV-2 reference strain sequences from Italy and Asia. The obtained sequence was submitted to the DDBJ/EMBL/GenBank databases under accession number ON677437.

To elucidate the genetic relationships of the analyzed CPV-2b strain with other CPV-2 strains, three phylogenetic trees, based on the complete coding sequence and full-length NS1 and VP2 genes, were constructed by IQ-TREE 2 [40]. The best model for distance estimates was used, identified as the one with the lowest Bayesian information criterion (BIC) by the ModelFinder function [41] (near complete genome: TPM3+F+R3; NS1: HKY+F+R2; VP2: TVM+F+R4), and with all *Carnivore protoparvovirus 1* sequences available in the GenBank database in March 2022 (near complete genome: 478; NS1: 645; VP2: 3988). The Ultrafast bootstrap approximation (ufBoot) [42] and SH-like approximate likelihood ratio test (SH-aLRT) [43] were used to assess branch robustness.

To evaluate the presence of potentially recombinant sequences, alignments were tested with all the different methods included in the RDP 5 software package [44]. 

## 5. Conclusions

The detection of this CPV-2 strain in a rescued stray litter represents the first evidence in Europe of a divergent Asian-like CPV-2b strain related to Asian CPV-2c-like strains and suggests its probable transmission within the local canine population. Further studies and large-scale surveys are necessary to evaluate CPV-2 evolution and the spread of this CPV-2b variant, and to assess its coexistence with genetically divergent strains in different geographical environments or ecological niches.

## Figures and Tables

**Figure 1 ijms-23-11240-f001:**
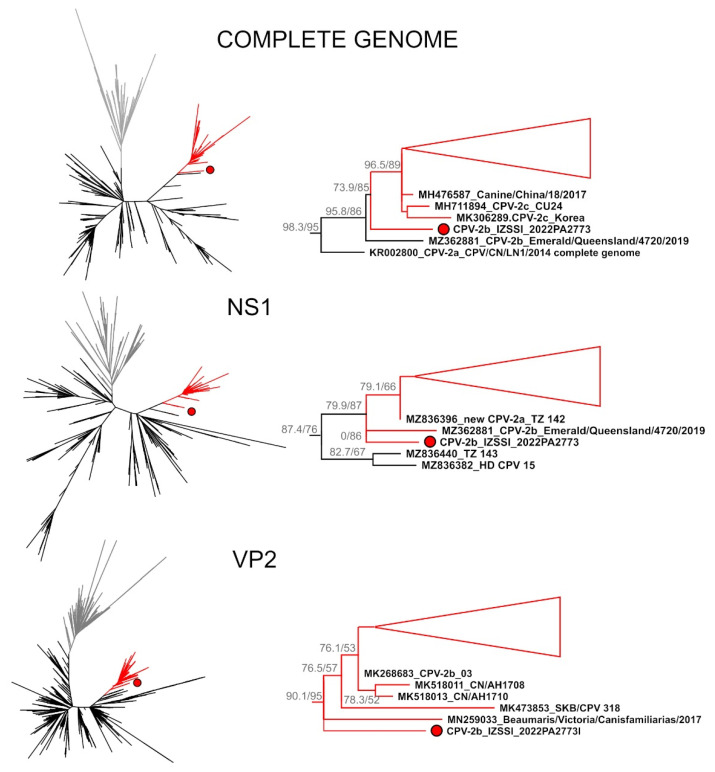
Phylogenetic analyses of the CPV-2b strain identified in this study. The phylogenetic trees compare the sequence obtained in this study, indicated by a red circle, with all CPV-2 and FPLV (used as outgroup and indicated by grey branches) sequences available in the GenBank. The full trees based on the complete genome (**top**), NS1 (**middle**) and VP2 (**bottom**) genes are shown on the left while enlargements of the clades, including the strain identified in this study (indicated by red branches and including Asian CPV-2c-like strains), are shown on the right. Triangles depict collapsed clades, which are shown in extenso in Appendix A. Trees were built with the maximum-likelihood method with IQ-Tree, and the outcomes of the SH-aLRT and bootstrap test (1000 replicates) are shown for the main nodes. Sequences are labelled with their accession number followed by the strain names.

**Table 1 ijms-23-11240-t001:** Amino acid variations in NS1 and VP2 sequences of analyzed CPV-2b strain.

CPV-2 Sequence					NS1						VP2					
Variant	Strain	Country	Year	Acc. Nr.	60	544	545	588	630	5	267	297	324	370	426	440
CPV-2 ^1^	CPV-b	USA	1978	M38245	I	Y	E	S	L	A	F	S	Y	Q	N	T
CPV-2a ^1^	43-91	Italy ^5^	1997	MF177224	-	F	-	-	-	-	-	A	-	-	N	A
CPV-2b ^1^	1-00	Italy ^5^	1999	MF177226	-	F	-	-	-	-	-	-	-	-	D	-
CPV-2b ^1,2^	IZSSI_PA18546/18	Italy ^5^	2018	MT981023	-	F	-	-	-	-	-	A	-	-	D	-
CPV-2c ^1^	485-09	Italy ^5^	2009	MF177228	-	-	-	-	-	-	-	A	-	-	E	-
CPV-2a ^1,3^	CPV/CN/LN1/2014	China ^6^	2014	KR002800	V	F	V	-	P	-	Y	A	I	-	N	A
CPV-2b ^1^	Canine/China/19/2017	China ^6^	2017	MH476588	-	-	-	-	-	-	Y	A	I	-	D	A
CPV-2c ^1^	Canine/China/12/2017	China ^6^	2017	MH476581	V	F	V	-	P	G	Y	A	I	R	E	-
**CPV-2b ^4^**	**CPV-2b_IZSSI_2022PA2773**	**Italy**	**2022**	**ON677437**	**V**	**F**	**V**	**N**	**P**	**G**	**Y**	**A**	**I**	**R**	**D**	**-**

^1^ Reference strains; ^2^ collected in the same region as the strain from this study; ^3^ strain belonging to Asian CPV-2c-like clade; ^4^ strain analyzed in this study; ^5^ European or ^6^ Asian CPV-2 reference strains; ‘-’ same amino acid as in the first row. The last row was bolded to highlight data and features of the analysed sequence.

## Data Availability

The sequence obtained in this study has been submitted to GenBank under accession number ON677437.

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
