# Peer review of "Identification and Molecular Characterization of a Divergent Asian-like Canine Parvovirus Type 2b (CPV-2b) Strain in Southern Italy"

_ijms, 2022, doi:10.3390/ijms231911240_

Round 1

Reviewer 1 Report

The manuscript entitled “Identification and Molecular Characterization of a Divergent Asian-like Canine Parvovirus Type 2b (CPV-2b) Strain in Southern Italy” describes the identification and molecular characterization of a canine parvovirus-2b (CPV-2b) strain from a stray puppy in Southern Italy. The CPV-2b strain has similarities to Asian CPV-2b strains based on amino acid changes in the NS1 and VP2 protein.
Monitoring the circulation of the CPV variants through genome analysis in different geographic areas is important to understand the virus evolution and is then a topic of interest. However, the conclusion of the manuscript is quite speculative, particularly because the results are limited to only one strain.

One main concern is related to the CPV classification. CPV-2a/b/c is an antigenic characterization that correlates with a single amino acid change. There have been attempts to achieve a genetic characterization of the strains, but it is still debatable. It is worth clarifying that the original CPV-2 type also has 426Asn and that "2a" can refer to the CPV-2a lineage that includes all antigenic variants, which are the ones that vary their antigenicity by changing the 426 residues. More important, changes in the 426 codon occur frequently and could alter the antigenic characterization while retaining the phylogenetic classification. Accordingly, the classification of these strains as CPV-2b is not necessarily relevant when analyzing the phylogenetic relationships and comparing the strains with 2a/2b and 2c strains could be misleading.

On the other hand, the classification as “new CPV-2b” based on Ala at residue 297 is strictly genetic (identify genogroups) but it does not offer any phylogenetic information for current strains. I would remove this classification from the manuscript.

Considering that the 2a/2b and 2c classification is antigenic and that the authors said that the geographic origin of the strain is uncertain, it is possible that the title was misleading and should be changed.

Additional points

The phylogenetic tree does not reveal that the CPV-2b strain (CPV-2b_IZSSI_2022PA2773) clusters with Asia and Australia strains. The authors called this cluster an “Asian” clade, which can be confusing for the readers.

Which was the criteria to select the strains in Table 1?

There are some English typos, some are included below:

19: “positive only for CPV-2”

66 “areas support the need”

91 “the nearly complete genome sequence”

252 “Extracted DNA/RNA was subjected”

234 “under positive selection pressure and convergent mutations”

217 “a heterogenous origin”

Author Response

Point-by-point response to Reviewer 1

Manuscript ID: ijms-1897345

Title: Identification and molecular characterization of a divergent Asian-like canine parvovirus type 2b (CPV-2b) strain in southern Italy

A: We thank the reviewer for taking the time to assess our manuscript. We have considered the Reviewer’s suggestions and revised the paper accordingly. Edits made to the manuscript are highlighted in yellow and below we provide the answers to each comment.

R: The manuscript entitled “Identification and Molecular Characterization of a Divergent Asian-like Canine Parvovirus Type 2b (CPV-2b) Strain in Southern Italy” describes the identification and molecular characterization of a canine parvovirus-2b (CPV-2b) strain from a stray puppy in Southern Italy. The CPV-2b strain has similarities to Asian CPV-2b strains based on amino acid changes in the NS1 and VP2 protein.

Monitoring the circulation of the CPV variants through genome analysis in different geographic areas is important to understand the virus evolution and is then a topic of interest. However, the conclusion of the manuscript is quite speculative, particularly because the results are limited to only one strain.

One main concern is related to the CPV classification. CPV-2a/b/c is an antigenic characterization that correlates with a single amino acid change. There have been attempts to achieve a genetic characterization of the strains, but it is still debatable. It is worth clarifying that the original CPV-2 type also has 426Asn and that "2a" can refer to the CPV-2a lineage that includes all antigenic variants, which are the ones that vary their antigenicity by changing the 426 residues.

A: We thank the reviewer for this comment. Indeed, the description of viral classification and evolution was not clear in the original version of the manuscript, and we have now clarified it according to the suggestions of the Reviewer (lines 44-47).

R: More important, changes in the 426 codon occur frequently and could alter the antigenic characterization while retaining the phylogenetic classification. Accordingly, the classification of these strains as CPV-2b is not necessarily relevant when analyzing the phylogenetic relationships and comparing the strains with 2a/2b and 2c strains could be misleading. On the other hand, the classification as “new CPV-2b” based on Ala at residue 297 is strictly genetic (identify genogroups) but it does not offer any phylogenetic information for current strains. I would remove this classification from the manuscript.

A: The Reviewer is right by stating that the antigenic classification of CPV-2 (either of the two positions mentioned by the Reviewer) does not correlate with viral phylogeny. The authors of this manuscript are well aware of this, as we already specify multiple times in the text at lines 54-57 (However, although possibly antigenically relevant [2], virus typing based on the analysis of these sole residues does not reflect viral evolution or the phylogenetic relationships between strains since viruses of the same antigenic type often do not cluster together [8,9]), lines 203-204 (Indeed, mutations at this site occur frequently and it has been shown that they have emerged multiple times throughout the evolutionary history of CPV-2 [8]), and lines 246-248 (This further highlights, as we previously observed [9], how a classification based on mutations at one single site is unreliable to study CPV-2 spread, but indicates that this could be possible by using a set of mutations throughout the whole genome.).

However, this antigenic classification system is widely used in literature, and it would not be fair from our side to ignore it. Additionally, the message of this manuscript was that this strain (antigenically classified as CPV-2b) possesses other antigenic and phylogenetic characteristics of CPV-2c strains, highlighting exactly how a single-amino acid classification is unreliable. The comparisons with other CPV-2a/b/c was performed precisely to show these incongruences. As we disclaim these interpretative limitations clearly in the manuscript, we are confident that our analyses or the presentation of our results prevent any misleading classification when analysing the phylogenetic relationships as, in this case, they only focus on the typing of this specific CPV-2 strain.

R: Considering that the 2a/2b and 2c classification is antigenic and that the authors said that the geographic origin of the strain is uncertain, it is possible that the title was misleading and should be changed.

A: While the specific area where this strain originated (i.e., where it acquired the mutations) is unknown, the strain clearly belongs to a monophyletic group of viruses with specific genetic traits that is commonly known in literature as “Asian CPV-2c”. Additionally, the virus is genetically and antigenically classified as CPV-2b, and it is divergent from all currently known CPV-2 strains. We believe that the title reflects the finding of our study and prefer to keep it as it is now.

Additional points

R: The phylogenetic tree does not reveal that the CPV-2b strain (CPV-2b_IZSSI_2022PA2773) clusters with Asia and Australia strains. The authors called this cluster an “Asian” clade, which can be confusing for the readers.
A: Figure 1 clearly shows that the strain identified in this study is close to viruses from Queensland and Victoria (Australia), as well as to viruses of Asian origins, although it does not form supported sub-clades with these viruses in any of the three trees. This is why we could not conclude that Australia, or some other specific geographic areas, is where this virus originated. Previous studies (Virus Evol. 2018 Apr 9;4(1):vey011. doi: 10.1093/ve/vey011; Transbound Emerg Dis. 2021 May;68(3):1445-1453. doi: 10.1111/tbed.13811), including those describing these strains from Australia (Viruses. 2021 Jun 6;13(6):1083. doi: 10.3390/v13061083; Transbound Emerg Dis. 2021 Mar;68(2):656-666), are convergent in the current presence of “Asian” and “Western” clades.

This Asian clade is a clade of viruses that originated in Asia but, since 2017, are distributed in other continents, suggesting their recent global spread. Therefore, the clade now includes viruses from all around the World, not only from Asia. There are several reports about this clade in literature and it is commonly known as “Asian CPV-2c”. We explain this clearly in the introduction at lines 62-67: “In the last decade, several studies analyzed the spread of CPV-2 in domestic and wild carnivorans in Italy [11,12] and documented the introduction and the rapid diffusion in the Italian canine population of CPV-2c strains of Asian origin (Asian CPV-2c-like strains) [13,14]. These strains, which have now been detected worldwide [15–17], are characterized by a specific set of mutations in both structural and non-structural proteins and are phylogenetically related to each-other.”

R: Which was the criteria to select the strains in Table 1?

A: We selected viral strains, which had both NS1 and VP2 complete gene sequences available, from Italy and from China to underline the divergence between the CPV-2a/2b strains circulating until now in Italy and the closer relationship of the new strain to Asian strains. The current availability of nearly complete CPV-2 genome sequences limited this selection, as underlined at lines 230-235 of the manuscript. This table was included to synthetically highlights the key amino acid variations, more extensively reported in Supplementary FigureS2. For greater clarity, we added a sentence to the text at line 271: “Key amino acid variations were deduced by comparing the analysed sequence with the whole dataset and synthetically depicted in table 1, by including available nearly complete CPV-2 reference strain sequences from Italy and from Asia.”.

There are some English typos, some are included below:

R: 19: “positive only for CPV-2”

A: This has been corrected.

R: 66 “areas support the need”

A: The verb “supports” is referred to “the occurrence” and it is, therefore, correct. 

R: 91 “the nearly complete genome sequence”

A: This has been corrected.

R: 252 “Extracted DNA/RNA was subjected”

A: This has been corrected.

R: 234 “under positive selection pressure and convergent mutations”

A: We have not changed this as “of convergent mutations” was part of “the presence of”: “the presence of sites under positive selection pressure and of convergent mutations”.

R: 217 “a heterogenous origin”

A: This has been corrected.

Author Response

Point-by-point response to Reviewer 2

Manuscript ID: ijms-1897345

Title: Identification and molecular characterization of a divergent Asian-like canine parvovirus type 2b (CPV-2b) strain in southern Italy

We thank the Reviewer for taking the time to assess our manuscript and for the positive recommendation.

Reviewer 3 Report

In the manuscript entitled “Identification and Molecular Characterization of a Divergent 2 Asian-like Canine Parvovirus Type 2b (CPV-2b) Strain in Southern Italy” authors have presented a case study of stray pups of a litter having gastroenteric signs and identified the involvement of the novel CPV. With further molecular characterization and phylogenetic study, the authors identified it as a novel CPV-2b strain which is related to Asian CPV-2c-like strains. In my opinion, the study was conducted at a fairly high level, the results look reliable and convincing.

Author Response

Point-by-point response to Reviewer 3

Manuscript ID: ijms-1897345

Title: Identification and molecular characterization of a divergent Asian-like canine parvovirus type 2b (CPV-2b) strain in southern Italy

We thank the Reviewer for taking the time to assess our manuscript and for the positive recommendation.

Round 2

Reviewer 1 Report

The manuscript has been modified and can be accepted in its present form.